# How Temporal Order of Inconsistent CSR Information Affects Consumer Perceptions?

Juhua Xu and Eun-Kyoung Han *


Department of Media and Communication, Sungkyunkwan University, Seoul 03063, Korea;
echo020716@skku.edu
* Correspondence: bird24@skku.edu

**Abstract:** What happens first between a corporate social responsibility (CSR) communication and a crisis can result in different levels of perceived cognitive dissonance and corporate hypocrisy depending on whether there is information inconsistency between the CSR communication and the crisis. This paper presents the findings from an experimental study and an online survey conducted and administered to investigate the contingency influence on consumer perceptions in response to inconsistent information. The results indicate that consumers experience greater cognitive dissonance and perceive more corporate hypocrisy when they are exposed, first, to a CSR initiative and then to a crisis, than when the order is reversed, provided that the CSR initiative and the crisis are congruent with the same social issue. However, there are no significant differences when the CSR initiative is incongruent with the crisis. Further, the findings of the study suggest that consumer cognitive dissonance not only directly influences the perceived corporate reputation, but also indirectly affects the perceived corporate reputation through a mediating effect of perceived corporate hypocrisy. The theoretical contribution of this study lies in providing a better understanding of consumer perceptions (including cognitive dissonance, perceived corporate hypocrisy and corporate reputation) in response to inconsistent CSR information. Meanwhile, the managerial contribution of this study stands by providing insights into the use of CSR communication strategies.

**Keywords:** temporal order; issue congruence; perceived corporate hypocrisy; cognitive dissonance; perceived corporate reputation

## 1. Introduction

Companies today commonly engage in corporate social responsibility (CSR) activities for a number of compelling reasons. One, consumers today place great emphasis on CSR when making purchase decisions [1]. Two, some scholars believe that CSR functions as a shield against adverse events such as crises [2,3]. Some researchers have also identified that CSR affects brand and company image [4–7], consumer attitude [5–7], purchase intention [4], and corporate reputation [5–7], as well as financial performance [4]. Other researchers have identified halo [8,9] or buffering [10] effects of pre-crisis CSR communication in the days following actual crises.

However, CSR has also been recognized as a double-edged sword, given that what companies actually do can be inconsistent with their CSR messaging. In particular, CSR communications can backfire when a company is involved in a crisis as it is initiating a CSR campaign in a situation that presents consumers with inconsistent information. Wagner et al. [1] first extended the concept of hypocrisy into the corporate domain and showed that inconsistency between CSR policies and practices could lead to consumer perceptions of corporate hypocrisy, which in turn, leads to negative CSR beliefs and unfavorable attitudes toward companies. In short, inconsistent CSR information plays a key role in perceived corporate hypocrisy.

Wagner et al. [1] further found that temporal order has an important influence on consumers' perceptions of corporate hypocrisy: The announcement of a CSR policy prior

to a reported crisis (proactive CSR strategy) generated higher perceptions of corporate hypocrisy than an announcement that followed a crisis did (reactive CSR strategy). That study's findings are of great significance, but Wagner et al. failed to consider the congruence effect; rather, they conducted their study under the assumption that the CSR policy and the crisis were both related to the same social issue. To clarify, it is important to examine if the same findings hold when a CSR policy and a crisis are incongruent with each other, that is, if issue congruence between a CSR initiative and a crisis functions as a moderator in the relationship between inconsistent CSR information and perceived corporate hypocrisy.

Additionally, it has long been established in psychology that inconsistent information leads to cognitive dissonance [11,12]. Given that information inconsistency between a CSR policy and a practice could generate perceptions of corporate hypocrisy, it is feasible to expect a potential relationship between cognitive dissonance and perceived corporate hypocrisy. Therefore, we introduced cognitive dissonance as a potential psychological variable in the relationship between inconsistent information and perceived corporate hypocrisy.

Overall, with this study, we set out to explore the moderating effects of issue congruence between a CSR initiative and a crisis in the link between the temporal order of a CSR initiative and a CSR crisis and perceived corporate hypocrisy. We also aim to examine if inconsistent CSR information could result in consumer cognitive dissonance and if perceived corporate hypocrisy is a mediating variable connecting consumer cognitive dissonance and perceived corporate reputation. The findings of this study theoretically contribute to a better understanding of consumer perceptions (including cognitive dissonance, perceived corporate hypocrisy, and corporate reputation) in the context of exposure to inconsistent CSR information. Additionally, this study offers insights into the managerial implications of using CSR as a communication strategy for practitioners.

## 2. Literature Review

### 2.1. CSR and Studies on Corporate Hypocrisy

CSR refers to a company's obligation to serve society positively [13], and companies' CSR efforts and messaging bring benefits in terms of corporate reputation. Companies that devote revenues to CSR activities do so publicly, and consumers reward socially responsible companies. According to extant studies, CSR can improve company image [14], and CSR perceptions influence consumer brand attitudes [15] and purchase behaviors [16]. Consumers prefer products from socially responsible companies and even seek job opportunities at them [17]. In short, CSR can be beneficial to companies.

However, although CSR brings benefits, it does carry drawbacks as well; one that has been capturing academic attention, is perceived corporate hypocrisy. Hypocrisy is the condition when a deviation exists between a person's assertions and the person's actions [18]. Wagner et al. first extended hypocrisy to the corporate domain and defined corporate hypocrisy as "the belief that a firm claims to be something that it is not" [1] (p. 49). The gap between the claims and the reality produces inconsistent information that leads consumers to perceive corporate hypocrisy. People perceive insincerity when companies fail in matching their behaviors with their advocated social responsibility standards [19].

Since Wagner et al.'s [1] first investigation, research has greatly expanded on corporate hypocrisy following both qualitative and quantitative methods. However, there are relatively few such studies, particularly quantitative studies. In one extant qualitative study on corporate hypocrisy, Fassin and Buelens [20] studied hypocrisy and sincerity on a continuum in the context that the business world is nuanced, not just black and white, and in another, Jauernig and Valentinov [21] introduced hypocrisy avoidance approach by considering a positive role of skepticism. Glozer and Morsing [22] aimed to answer the question of whether hypocrisy is always undesirable, and Wagner et al. conducted a later study [23], in which they furthered the conceptualization of corporate hypocrisy by classifying it as moral or behavioral.

In contrast with the above works, quantitative research on corporate hypocrisy has mainly focused on measuring the construct as well as uncovering its antecedents and negative consequences. Goswami et al. [24] developed a nine-item scale for measuring employees' perceptions of corporate hypocrisy. Guèvremont [25] developed a 12-item scale for measuring brand hypocrisy, which they divided into the categories of image, mission, message, and social hypocrisy; admittedly, however, brand hypocrisy is different from corporate hypocrisy. Studies have also shown that perceived corporate hypocrisy has a negative effect on corporate reputation via mediating effects of perceived CSR [26], worsens consumer attitudes toward companies [1,26,27], and leads to counterproductive work behaviors against employers through a mediating effect of organizational identification.

In terms of antecedents of corporate hypocrisy, Wagner et al. [1] explored how proactive (CSR statement followed by CSR violating behavior) and reactive (CSR behavioral violation followed by CSR statement) CSR strategies differed in influencing consumer perceptions of corporate hypocrisy and attitudes toward a firm. Shim and Yang [24] proposed a model that encompassed corporate reputation, the occurrence of a crisis, and CSR history as determinants of corporate hypocrisy. Additionally, Wang and Zhu [28] proposed that internal attribution of corporate hypocrisy contributes to consumer perception of corporate hypocrisy. Kim et al. [29] introduced corporate brand trust as a mediator between consumer CSR perception and perceived corporate hypocrisy, while Wagner et al. [23] identified that key antecedents of corporate hypocrisy include valence, order, information source, and authenticity [1,30–32].

As presented above, investigating corporate hypocrisy can follow a number of different paths; however, our first goal was to contribute to a better understanding of its uncovered antecedents. Specifically, we focused on furthering Wagner et al.'s [1] findings by introducing a potential situational variable. In addition, our study is characterized by encompassing additional consumer psychological process variables in order to more comprehensively explain the psychological mechanisms that operate when consumers are exposed to inconsistent CSR information.

*2.2. Effects of Temporal Order of a CSR Initiative and a Crisis on Consumer Perception of Corporate Hypocrisy with Issue Congruence as a Moderator*

Hypocrisy, whether individual or corporate, involves two behaviors: an assertion and an action [1,18]. Furthermore, any examinations of the concept inevitably encompass determining which occurred first and whether their order has any impacts on outcomes, that is, establishing order effects. Extant studies have shown that temporal order matters in influencing perceptions; researchers have identified both primacy effects, whereby people tend to better remember information acquired earlier, and recency effects that entail better remembering information received last [1]. One study on temporal order that is closely related to our study is that by Barden et al. [33], although these authors studied people's judgments of hypocrisy in other individuals rather than in companies. In three sub-studies, Barden et al. consistently found that people perceived greater hypocrisy in others when a statement was followed by a behavior violation than when the order was reversed.

As noted earlier, in the first examination of corporate hypocrisy as a construct, Wagner et al. [1] identified that a proactive CSR communication strategy, in which a company's CSR announcement was followed by a negative CSR behavior resulted in perceptions of greater corporate hypocrisy than when a company's negative CSR behavior preceded the CSR announcement. However, in practice, companies do use proactive CSR communication strategies to their benefit because these strategies improve company image [14], gain positive reactions from consumers [5], and even function as shields against attacks of negative events in the future. In comparison, reactive CSR communication strategies are more often used in response to crises and negative events to minimize the damage incurred [34].

The combined findings from academic studies and marketing practices suggest a number of explanations. In one possibility, when the CSR assertion precedes an opposite action, the announcement creates a higher expectation for the company's actions, and this

increases the disappointment and perceptions of hypocrisy in reaction to the subject's behavioral violation; this possibility follows the Chinese proverb "the greater the expectation, the greater the disappointment". In another possibility, the contradictory behavior occurs before the assertion, and the announcement of the subject's effort to remedy the behavior creates hope for improvement. This situation generates a lower perception of hypocrisy and results in less negative consequences in general.

However, Wagner et al. [1] conducted their study under the assumption that a CSR announcement and negative behavior were both related to the same social issue, and they did not consider perceptions of corporate hypocrisy when a CSR initiative and a crisis are associated with two different social issues. To close that gap in the literature, we posed the question of whether Wagner et al.'s findings would hold when the CSR initiative was incongruent with the crisis. In this research stream, issue congruence between a CSR initiative and a crisis is defined as "the degree of fit between a crisis issue and the CSR initiative" [10] p. 448. One example of a proactive CSR strategy with an incongruent issue is as follows: An apparel company claims that it is devoted to supporting animal protection by donating a portion of their revenue to the World Wide Fund, but media later report that the company employs child labor in Bangladesh to manufacture its garments. The converse scenario is a reactive CSR strategy under which the media report that this apparel company hires child labor in Bangladesh to manufacture their garments and then the company announces that it is going to support animal protection by donating a portion of its revenue to the World Wide Fund. In both of these scenarios, the company's CSR initiative is incongruent with the crisis that occurs (animal protection versus human rights violations), and the question in this context is whether a proactive CSR strategy still leads to more perceived corporate hypocrisy than a reactive strategy does?

When a company launches a CSR initiative but later experiences a crisis in a different domain (in other words, a proactive CSR strategy in an incongruent situation) such as with the apparel company example, consumers might be more understanding of the situation and think less negatively of the company because the unrelated crisis in another domain does not reflect a violation of the company's CSR initiative. However, whether consumers are more or less forgiving, a crisis is still a crisis, and some consumers will still have negative thoughts about the company. We suggested that with an incongruent issue, consumers will continue to perceive hypocrisy in the company but at a lower level, whereas when the crisis of an alleged socially irresponsible behavior, consumers will expect the company to adopt an appropriate and immediate response strategy to remedy the situation [35]. However, a reactive response in an incongruent situation is to initiate a CSR strategy in an unrelated domain. According to Kim and Choi [10], if the company's behavior is not in line with consumers' expectations, consumers will elaborate more of the company's CSR efforts, potentially with more suspicion of the company's motives. In short, we expected that consumers would perceive a company as more hypocritical when instituting a reactive CSR strategy than they would with a company that followed a proactive CSR strategy in an incongruent situation. Based on the analysis above, we proposed the following hypotheses:

**Hypothesis 1 (H1).** *Issue congruence between a CSR initiative and a CSR crisis moderates the effects of the temporal order of the initiative and the crisis on consumer perception of corporate hypocrisy.*

**Hypothesis 1 (H1a).** *The temporal order of a CSR initiative and a CSR crisis has an effect on consumer perception of corporate hypocrisy when the initiative and the crisis are in the same domain. Specifically, a proactive CSR strategy (initiative followed by crisis) will lead to a higher perception of corporate hypocrisy than will a reactive CSR strategy (initiative follows crisis) when the CSR initiative and the CSR crisis are congruent with each other.*

**Hypothesis 1 (H1b).** *The temporal order of a CSR initiative and a CSR crisis has an effect on consumer perception of corporate hypocrisy when the initiative and the crisis are not in the same domain. Specifically, a proactive CSR strategy (initiative followed by crisis) will lead to a lower*

*perception of corporate hypocrisy than will a reactive CSR strategy (initiative follows crisis) when the CSR initiative and the CSR crisis are incongruent with each other.*

### 2.3. Effects of Temporal Order of a CSR Initiative and a CSR Crisis on Consumer Cognitive Dissonance with Issue Congruence as a Moderator

While the consequences of temporal order in the work by Wagner et al. [1] directly led to consumer perception of corporate hypocrisy, we considered that consumer perception is a complicated cognitive mechanism and that more cognitive process variables are involved in attitude formation. Thus, we also proposed that the temporal order of a CSR initiative and a crisis has a direct effect on consumer cognitive dissonance. Cognitive dissonance theory holds that when exposed to an imbalanced state of cognition, attitudes, and behaviors, people will reach a psychological discomfort state and experience tensions and that they will be motivated to eliminate this discomfort through reconciliatory attitude change to reach the state of cognitive balance [11,12].

Scholars have identified two types of dissonance, one that related to making a purchase decision and the emotional dissonance, and one that involved with knowledge and cognitions about oneself [36,37]. Earlier researchers applied cognitive dissonance theory in the context of before and after consumers make purchase decisions. Before making purchase decisions, multiple alternatives can create cognitive discomfort [36,38,39], and in one study, the discomfort was closely tied to the product itself. Consumers in that study felt more discomfort when they were more involved with the product [40]. In another study, the researchers concluded that the intensity of consumers' cognitive dissonance depended on the importance of the decision, the number of alternatives and their attractiveness, and the similarities among them [41]. Post-purchase cognitive dissonance results when a product's performance fails to match consumers' pre-purchase expectations, which leads to consumer mental discomfort [42,43].

Another stream of studies on cognitive dissonance related to self-knowledge and cognitions involves inducing cognitive dissonance and then generating attitudinal and/or behavioral change. The induced compliance paradigm can induce cognitive dissonance by asking people to advocate certain behavior that is contradictory with their attitudes, and the hypocrisy paradigm can cause dissonance by making people recall past experiences of not behaving in the ways they advocate [37]. Gharib [44] induced cognitive dissonance by asking participants to write essays about the benefits of not eating meat, which contradicted their attitudes. Sharifi and Esfidani [37] induced cognitive dissonance by making participants realize that they had not participated actively in religious behaviors that they had advocated, and they found that the dissonance had in turn induced emotions such as guilt and shame.

Based on our review of the literature, we determined that studies on cognitive dissonance are abundant but that the traditional study subjects have been "insiders" to the relevant cases; participants have been the product purchasers or the actors whose actions did not match their advocated behaviors. In contrast, we focused on exploring cognitive dissonance in consumers as outsiders, exposing them to two inconsistent pieces of information about a third party, a company, rather than the consumers themselves. Our assumption was that exposing consumers to a corporate CSR initiative and a CSR crisis would induce cognitive dissonance in them in response to the two inconsistent pieces of information. Specifically, we postulated that when a company initiated a proactive CSR strategy and then experienced a crisis with the same social issue, consumers would form initial positive impressions of the company based on its CSR stance but that these attitudes would shift to negative to resolve the cognitive distress a crisis involving the company. In contrast, we proposed that if a company were involved in a crisis first and then announced its CSR strategy as a response to the crisis, consumers would experience less intense cognitive distress because the company was trying to correct its wrongdoing; this outcome is consistent with common sense.

However, in the event that the company advocated a proactive CSR strategy and then later experienced an unrelated crisis as opposed to experiencing the crisis first and then

announcing an unrelated CSR strategy, consumers would feel less cognitive discomfort in the former situation because the company's behavior was not contradicting its previous statements, but their discomfort would be greater in the latter scenario because consumers would expect the company to correct what it did wrong and would be disappointed at the irrelevant statement. Following this stream of analysis, we proposed the following hypotheses:

**Hypothesis 2 (H2).** *Issue congruence between a CSR initiative and a CSR crisis moderates the effects of the temporal order of the initiative and the crisis on consumer cognitive dissonance.*

**Hypothesis 2 (H2a).** *The temporal order of a CSR initiative and a CSR crisis has an effect on consumer cognitive dissonance when the initiative and the crisis are in the same domain. Specifically, a proactive CSR strategy (initiative followed by crisis) will lead to higher consumer cognitive dissonance than will a reactive CSR strategy (initiative follows crisis) when the initiative and the crisis are congruent with each other.*

**Hypothesis 2 (H2b).** *The temporal order of a CSR initiative and a CSR crisis has an effect on consumer cognitive dissonance when the initiative and the crisis are not in the same domain. Specifically, a proactive CSR strategy (initiative followed by crisis) will lead to lower consumer cognitive dissonance than a reactive CSR strategy (initiative follows crisis) when the initiative and the crisis are incongruent with each other.*

*2.4. Relationship between Perceived Corporate Hypocrisy and Consumer Cognitive Dissonance*

As extant study findings demonstrate, there is a close connection between hypocrisy and cognitive dissonance. The hypocrisy paradigm [45,46] is frequently incorporated in experimental studies to induce cognitive dissonance. In the hypocrisy paradigm, cognitive dissonance is induced by asking study participants to recall a past behavior of violating their beliefs, creating a discrepancy between the behavior and a belief. Aronson et al. [45] applied the paradigm in asking participants to recall a past failure of not using condoms, even though they agreed that using condoms could prevent the spread of AIDS. Additionally, Fried [47] induced dissonance by asking participants to recall past transgressions of recycling behavior. In another study, Yousaf and Gobet [46] explored how religious hypocrisy induced by a discrepancy between religious attitude and behavior affected attitudes and emotions, and other researchers have applied the hypocrisy paradigm to create dissonance in participants to promote pro-environmental behaviors [48] and a meatless lifestyle [44]. Stone and Fernandez [49] argued that the success of the hypocrisy paradigm in influencing behaviors depended on the salience of the discrepancy between a social norm and a transgression.

Although all of these study findings indicate a close relationship between hypocrisy and cognitive dissonance, all of these previous researchers focused on manipulating experimental conditions in which participants experienced a discrepancy between their advocated beliefs and behaviors and assumed that the discrepancies reflected feelings of hypocrisy. That is, researchers have manipulated discrepancy, but none have actually measured hypocrisy and cognitive dissonance or explored their causal relationship. Realizing this gap, we acknowledge hypocrisy and cognitive dissonance as two different variables, apply them both to a corporate CSR context, and explore the relationship between them.

As suggested by Wagner et al. [1], measuring psychological mechanisms should incorporate rather complicated process variables rather than the simple stimulus-response reactions that were studied in earlier communications research. Instead, we suggest that before they perceive corporate hypocrisy in the face of a discrepancy between two pieces of contradictory information about a company, consumers will likely experience cognitive dissonance first and that it is this cognitive dissonance that leads consumers to be suspicious of companies' initiating CSR strategies and to believe companies are hypocritical. Based on the conclusions of earlier researchers that "any skepticism generated has the potential

to have broad negative consequences on a firm's reputation" [50,51], we proposed the following hypothesis:

**Hypothesis 3 (H3).** *Cognitive dissonance has a positive effect on perceived corporate hypocrisy.*

*2.5. Effects of Cognitive Dissonance and Perceived Corporate Hypocrisy on Corporate Reputation*

　　Corporate reputation is defined as "a cognitive representation of a company's actions and results that crystallizes the firm's ability to deliver valued outcomes to its stakeholders" [27,52] p. 69. It reflects how stakeholders evaluate a company [53], serving as "a collective representation of a past performance by multiple stakeholders who assess the company's ability" [29] p. 3686. In the context of this study, companies typically engage in CSR activities to improve their images and to establish favorable reputations [54,55], and corporate hypocrisy is detrimental to how consumers view companies [56]. Additionally, in the context of this study, it has been well established that cognitive dissonance influences attitude [11,12]. Therefore, we hypothesized that both cognitive dissonance and perceived corporate hypocrisy would influence consumer evaluations of a company such that the more consumers perceive corporate hypocrisy and the greater their cognitive dissonance is, the more poorly consumers will evaluate the company. We further postulated that perceived corporate hypocrisy would act as a mediator between cognitive dissonance and corporate reputation. Hence, we proposed the following hypotheses:

**Hypothesis 4 (H4).** *Consumer cognitive dissonance has a negative effect on perceived corporate reputation.*

**Hypothesis 5 (H5).** *Perceived corporate hypocrisy has a negative effect on corporate reputation.*

**Hypothesis 6 (H6).** *Perceived corporate hypocrisy mediates the relationship between consumer cognitive dissonance and corporate reputation.*

### 3. Research Methods

*3.1. Data Collection*

　　To test our proposed hypotheses, we employed a 2 (temporal order of a CSR initiative and a crisis: proactive vs. reactive) × 2 (issue congruence between a CSR initiative and a crisis: congruent vs. incongruent) between-subject full-factorial experimental design. A commercial online survey company, Wenjuanxing, was hired to conduct the survey with participants who were randomly assigned to one of four conditions. For the experiments, we created a mock company, Company H, to avoid any preexisting participant attitudes associated with any actual companies.

　　We described Company H as a multinational clothing retail company that was well-known for fast-fashion products, that operated in retail and manufacturing businesses around the world, and that also took on an active role in society. The reason we chose a clothing company was that clothes are relevant to everyone, irrespective of sex, age, education level, etc., and thus, it could apply to any population we sampled.

　　We created four hypothetical scenarios for Company H to reflect the manipulated temporal order and the issue congruence in the experimental conditions; the CSR initiative was consistent across all four scenarios. Specifically, Company H's CSR initiative reflected a commitment to environmental protection via textile recycling. The crisis in the congruent issue condition related to causing environmental pollution by burning unsold products and recycling textiles reflected the congruent manipulation, while in the incongruent issue manipulation, the crisis was associated with paying employees less than the government's minimum hourly wage. To manipulate temporal order, we organized the order of the CSR initiative and the CSR crisis to reflect either a proactive (CSR initiative first) or a reactive (crisis first) CSR strategy, with a time lag of two weeks between the two events, in keeping with extant studies [1,33]. A detailed description of the four scenarios can be found in Appendix A.

The online questionnaire opened with general information about Company H, and after reading the initial information, participants read additional information about the company, including its CSR initiative and crisis, with the order depending on whether a proactive or reactive CSR strategy was being presented. After reading all information, participants were first asked to answer manipulation check questions and then to use 7-point Likert scales to record measures of cognitive dissonance, corporate hypocrisy, and corporate reputation; the survey ended with demographic questions.

We hired the same online survey company to conduct a pretest on 82 participants to make sure that potential respondents would understand the four scenarios as we intended. For the pretest, participants first read the general description of Company H, its CSR initiative, and the crisis; they then answered questions about temporal order. Specifically, participants chose "before" or "after" in response to statements such as "Company H's incident of burning unsold clothes happened before or after its post of commitment to environment protection." Then, they answered five questions related to issue congruence between the CSR initiative and the crisis issue [10,57] on a 7-point Likert scale ranging from 1 (strongly disagree) to 7 (strongly agree):

1. "I think company H's incident of burning down unsold clothes and its post of commitment to environment protection are related to same social issue"
2. "Company H's incident of burning down unsold clothes relates to its post of commitment to environment protection"
3. "Company H's incident of burning down unsold clothes is a fit with company H's post of commitment to environment protection"
4. "Company H's incident of burning down unsold clothes is similar with company H's post of commitment to environment protection"
5. "Company H's incident of burning down unsold clothes represents a good match with company H's post of commitment to environment protection".

The pretest results showed that participants were able to identify the temporal order of a CSR initiative and a CSR crisis and also confirmed that manipulations of the issue congruence between the initiative and the crisis ($M_{congruence}$ = 5.280, SD = 1.117; $M_{incongruence}$ = 2.733, SD = 0.791; t(80) = 11.963, $p < 0.001$) were successful.

The same online survey company collected the data for the main experiment; a total of 242 responses. We used 239 responses in the final analysis, after we excluded surveys from three respondents who failed to correctly identify the temporal order. Sample descriptive analysis on demographic information showed that 52.3% of the respondents were male, with the majority (87.8%) aged 20 to 39 years old; for the remainder, 1.7% were under 19 years old, 9.6% were 40–49 years old, and 0.8% were over 50 years old. Over half of respondents, 61.1%, had a four-year university education or above, and the majority, 91.2%, had monthly salaries of less than 10,000 RMB.

*3.2. Measures*

3.2.1. Cognitive Dissonance

We measured cognitive dissonance with a three-item scale [37,58–60] that asked respondents to indicate how much they agreed with three items: "Overall, I am dissatisfied with company H", "After reading company H's environment pollution issue, I thought I'd been fooled by its environment protection announcement", and "I believe that company H is unable to meet my expectations". Respondents rated the items on a 7-point Likert scale ranging from 1 (*strongly disagree*) to 7 (*strongly agree*) (M = 4.859, SD = 1.114, α = 0.784), and we used the means for each item in the analysis.

3.2.2. Perceived Corporate Hypocrisy

We measured perceived corporate hypocrisy with the three-item scale developed by Wagner et al. [1]. Respondents were asked to indicate how much they agreed with these statements: "Company H acts hypocritically", "What Company H says and does are two different things", and "Company H pretends to be something that is not" on a 7-point

Likert scale, ranging from 1 (*strongly disagree*) to 7 (*strongly agree*) (M = 4.908, SD = 1.089, α = 0.795). We used the means of these three items in the analysis.

### 3.2.3. Corporate Reputation

We had the respondents rate corporate reputation with three items [61,62]: "Company H has a reputation for being honest", "Company H has a reputation for being reliable", and "Company H has a reputation for being trustworthy". These items were also rated on a 7-point Likert scale (1 *strongly disagree*, 7 = *strongly agree*) (M = 3.264, SD = 1.374, α = 0.829), and we used the means of these items in the analysis.

## 4. Results

### 4.1. Manipulation Check

An independent-samples $t$-test on issue congruence between the CSR initiative and the crisis revealed a significant difference between the congruence and incongruence conditions ($M_{congruence}$ = 5.428, SD = 0.945; $M_{incongruence}$ = 2.817, SD = 0.962; t (237) = 21.168, $p < 0.001$).

### 4.2. Hypothesis Testing

Pearson correlation analysis was first performed on three independent variables, and then MANOVA was performed to test H1, which predicted an interaction effect between temporal order and issue congruence on corporate hypocrisy, and H2, which predicted an interaction effect on cognitive dissonance. We performed MANOVA because, as Table 1 presents, there was a significant correlation between cognitive dissonance and corporate hypocrisy.

**Table 1.** Correlation matrix.

| Variables | M | SD | α | 1 | 2 | 3 |
|---|---|---|---|---|---|---|
| 1. Cognitive dissonance | 4.859 | 1.114 | 0.784 | 1 | | |
| 2. Corporate hypocrisy | 4.908 | 1.089 | 0.795 | 0.476 ** | 1 | |
| 3. Corporate reputation | 3.264 | 1.374 | 0.829 | −0.344 ** | −0.373 ** | 1 |

Note: n = 239, ** $p < 0.01$.

MANOVA revealed statistically significant results for a main effect of temporal order of the CSR initiative and the CSR crisis (Pilla's Trace = 0.049, F(2, 234) = 6.025, $p < 0.01$, partial $\eta^2$ = 0.049; Wilks' Lambda = 0.951, F(2, 234) = 6.025, $p < 0.01$, partial $\eta^2$ = 0.049; Hotelling's Trace = 0.051, F(2, 234) = 6.025, $p < 0.01$, partial $\eta^2$ = 0.049), a main effect of issue congruence between the initiative and the crisis (Pilla's Trace = 0.073, F(2, 234) = 9.210, $p < 0.001$, partial $\eta^2$ = 0.073; Wilks' Lambda = 0.927, F(2, 234) = 9.210, $p < 0.001$, partial $\eta^2$ = 0.073; Hotelling's Trace = 0.079, F(2, 234) = 9.210, $p < 0.001$, partial $\eta^2$ = 0.073), and an interaction effect between temporal order and issue congruence (Pilla's Trace = 0.137, F(2, 234) = 18.599, $p < 0.001$, partial $\eta^2$ = 0.137; Wilks' Lambda = 0.863, F(2, 234) = 18.599, $p < 0.001$, partial $\eta^2$ = 0.137; Hotelling's Trace = 0.159, F(2, 234) = 18.599, $p < 0.001$, partial $\eta^2$ = 0.137). MANOVA further revealed that temporal order significantly affected cognitive dissonance (F(1, 235) = 10.415, $p < 0.01$, partial $\eta^2$ = 0.042) and perceived corporate hypocrisy (F(1, 235) = 5.793, $p < 0.05$, partial $\eta^2$ = 0.024), issue congruence significantly affected cognitive dissonance (F(1, 235) = 13.324, $p < 0.001$, partial $\eta^2$ = 0.054) and perceived corporate hypocrisy (F(1, 235) = 12.043, $p < 0.01$, partial $\eta^2$ = 0.049), and the interaction effect between temporal order and issue congruence significantly affected cognitive dissonance (F(1, 235) = 27.416, $p < 0.001$, partial $\eta^2$ = 0.104) and perceived corporate hypocrisy (F(1, 235) = 23.773, $p < 0.001$, partial $\eta^2$ = 0.092). We confirmed an interaction effect between temporal order and issue congruence on cognitive dissonance and perceived corporate hypocrisy, and H1 and H2 were supported.

As Table 2 indicates, MANOVA results also showed that the proactive CSR strategy led to significantly higher perceptions of corporate hypocrisy when the CSR initiative and the crisis were congruent than the reactive CSR strategy did ($M_{proactive}$ = 5.607, $SD_{proactive}$

= 0.129; $M_{reactive}$ = 4.656, $SD_{reactive}$ = 0.130; F(1, 235) = 26.858, *p* < 0.001, partial $\eta^2$ = 0.103). However, there was no statistically significant difference between the effects of the proactive and reactive strategies on corporate hypocrisy when the CSR initiative and the CSR crisis were incongruent ($M_{proactive}$ = 4.517, $SD_{proactive}$ = 0.130; $M_{reactive}$ = 4.839, $SD_{reactive}$ = 0.133; F(1, 235) = 3.010, *p* > 0.05, partial $\eta^2$ = 0.013). Therefore, H1a was supported but H1b was not. MANOVA results also revealed that the proactive CSR strategy led to significantly higher cognitive dissonance than the reactive strategy when the CSR initiative and the crisis issue were congruent did ($M_{proactive}$ = 5.650, $SD_{proactive}$ = 0.130; $M_{reactive}$ = 4.539, $SD_{reactive}$ = 0.131; F(1, 235) = 36.272, *p* < 0.001, partial $\eta^2$ = 0.134). However, there was no statistically significant difference between the two strategies' effects on cognitive dissonance when the CSR initiative and the crisis were incongruent ($M_{proactive}$ = 4.483, $SD_{proactive}$ = 0.131; $M_{reactive}$ = 4.747, $SD_{reactive}$ = 0.133; F(1, 235) = 1.992, *p* > 0.05, partial $\eta^2$ = 0.008). Therefore, H2a was supported while H2b was not. Figures 1 and 2 illustrate these interaction effects.

**Table 2.** Means (SD) and *p* values.

| Experimental Condition | | Cognitive Dissonance | Corporate Hypocrisy |
|---|---|---|---|
| Congruence | Temporal Order | | |
| Congruent | Proactive | 5.650 (0.130) | 5.607 (0.129) |
| Congruent | Reactive | 4.539 (0.131) | 4.656 (0.130) |
| Incongruent | Proactive | 4.483 (0.131) | 4.517 (0.130) |
| Incongruent | Reactive | 4.747 (0.133) | 4.839 (0.133) |
| | | p (PES) | p (PES) |
| Temporal order | | 0.001 (0.042) | 0.017 (0.024) |
| Congruence | | 0.000 (0.054) | 0.001 (0.049) |
| Temporal order × Congruence | | 0.000 (0.104) | 0.000 (0.092) |

Note: PES, partial eta squared.

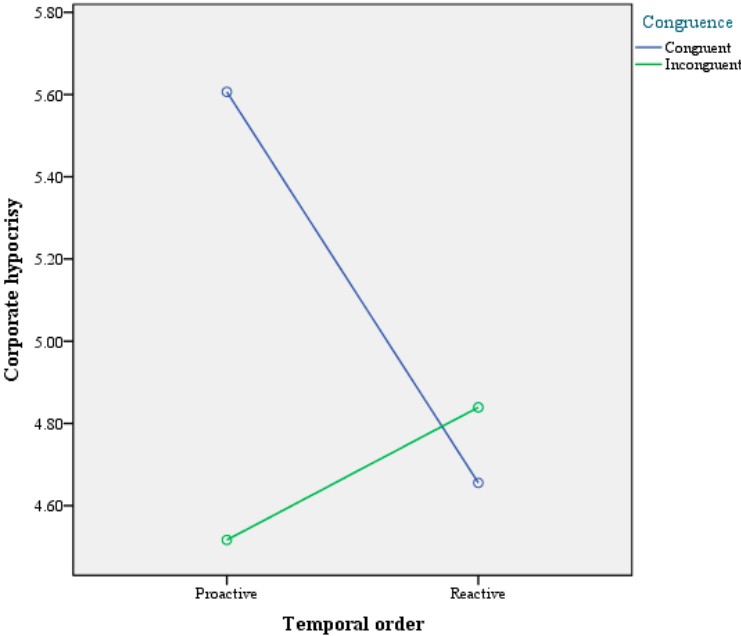

**Figure 1.** The interaction effect between temporal order and issue congruence on corporate hypocrisy.

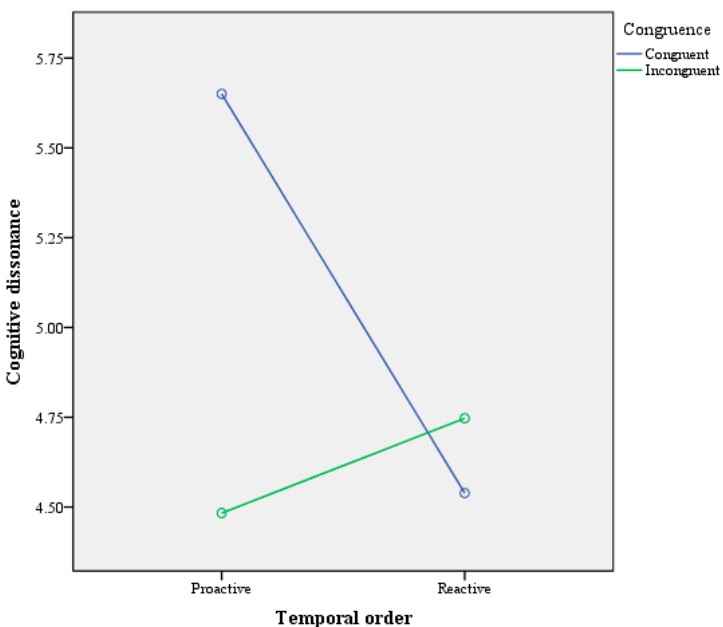

**Figure 2.** The interaction effect between temporal order and issue congruence on cognitive dissonance.

H3, H4, H5, and H6 together predicted direct and indirect mediation effects of corporate hypocrisy on the relationship between cognitive dissonance and corporate reputation. To test these hypotheses, we performed PROCESS macro model 4 [63]. As shown in Table 3, the results revealed that cognitive dissonance had a significant effect on corporate reputation ($\beta = -0.2651$, SE = 0.0829, $p < 0.01$). The results further showed that cognitive dissonance had a significant effect on corporate hypocrisy ($\beta = 0.4650$, SE = 0.558, $p < 0.001$), and corporate hypocrisy's effect on corporate reputation was also significant ($\beta = -0.3410$, SE = 0.0848, $p < 0.001$). Therefore, these results combined indicate that cognitive dissonance had both a direct negative effect on corporate reputation (CI 95% 1000 bootstrapped samples ($-0.4284$, $-0.1017$)) and an indirect negative effect on corporate reputation through corporate hypocrisy (CI 95% 1000 bootstrapped samples ($-0.2499$, $-0.0736$)). Therefore, H3, H4, H5, and H6 were all supported.

**Table 3.** Direct, indirect, and total effects.

| Effect. | Path | β | SE | t | LLCI | ULCI |
|---------|------|-----|-----|-----|------|------|
| Direct effect | Cognitive dissonance → Corporate reputation | −0.2651 | 0.0829 | −3.1972 ** | −0.4284 | −0.1017 |
| Indirect effect | Cognitive dissonance → Corporate hypocrisy | 0.4650 | 0.0558 | 8.3266 *** | 0.3550 | 0.5750 |
| | Corporate hypocrisy →Corporate reputation | −0.3410 | 0.0848 | −4.0206 *** | −0.5082 | −0.1739 |
| | Cognitive dissonance → Corporate hypocrisy → Corporate reputation | | | | −0.2499 | −0.0736 |
| Total effect | Cognitive dissonance → Corporate reputation | −0.4237 | 0.0752 | −5.6321 *** | −0.5719 | −0.2755 |

Note: LL = lower level, UL = upper level, CI = 95% confidence interval. Bootstrapped at 1000 samples. *** $p < 0.001$, ** $p < 0.01$.

Additionally, we performed MANOVA on cognitive dissonance and corporate hypocrisy with gender, education, and income, respectively. Results from MANOVA revealed that gender had no significant effect on neither cognitive dissonance (F(1, 237) = 0.051, $p = 0.822$) nor corporate hypocrisy (F(1, 237) = 0.286, $p = 0.593$). Neither did education on cognitive dissonance (F(3, 235) = 1.338, $p = 0.263$) or corporate hypocrisy (F(3, 235) = 0.299, $p = 0.826$),

nor income did on cognitive dissonance ($F_{(4, 234)} = 0.400$, $p = 0.808$) or corporate hypocrisy ($F_{(4, 234)} = 0.848$, $p = 0.496$).

## 5. Conclusions

With this study, we examined how the temporal order of a CSR initiative and a CSR crisis and issue congruence between the CSR initiative and the crisis interacted with each other to influence consumer cognitive dissonance and perceived corporate hypocrisy, as well as whether these perceptions further affected the corporate reputation. Results from the collected sample data first supported the existence of an interaction effect between temporal order and issue congruence. Specifically, we found that consumers experienced greater cognitive dissonance and perceived greater corporate hypocrisy when a company was involved in a crisis after it had initiated a CSR strategy that was in the same domain as the crisis issue than a reversed order. However, when the CSR initiative and the crisis were related to different social issues, there were no significant differences in the effects on consumer cognitive dissonance or perceived corporate hypocrisy in terms of whether the crisis occurred before or after the CSR initiative was announced. This finding rejected our prediction that a reactive CSR strategy would lead to higher consumer cognitive dissonance and greater perceived corporate hypocrisy than a proactive CSR strategy would when the CSR initiative and crisis were incongruent. In other study findings, consumer cognitive dissonance had both a direct effect on corporate reputation and an indirect effect through perceived corporate hypocrisy. We also found that demographic variables such as gender, education, and income had no effects on consumer cognitive dissonance or perceived corporate hypocrisy.

### 5.1. Theoretical and Managerial Implications

The distinctiveness of this study is that we combined the effects of a CSR initiative and a social responsibility crisis and examined their influence on consumer perceptions of a corporation using other consumer psychological process variables in addition to perceived corporate hypocrisy. The study makes a number of theoretical contributions.

First, our research supplements extant studies on corporate hypocrisy by establishing a moderating effect of issue congruence between a CSR initiative and a CSR crisis. Wagner et al. [1] showed that inconsistent information between the two led to consumer perception of corporate hypocrisy, and this perception was stronger when the company initiated its CSR strategy first and then experienced the crisis, even though consumers still considered a company hypocritical in a reversed order. However, these findings were only confirmed in our study when the CSR campaign and the company's crisis were related to the same social issue. When a company experiences a crisis that is unrelated to its CSR announcement, consumers still perceive the company as hypocritical because of the inconsistent information between the announcement and the crisis, but there was no significant difference related to whether the CSR initiative preceded or followed the crisis. Therefore, even though no significant difference was found in the incongruent condition, our study furthered Wagner et al.'s study by establishing issue congruence as a situational variable in the formation of corporate hypocrisy perception following inconsistent CSR information.

Second, we introduced the concept of cognitive dissonance into the corporate hypocrisy domain as extant studies have established that inconsistent information could lead to perceptions of dissonance [11,12]. Our research first confirms that consumers do experience cognitive dissonance following exposure to inconsistent CSR information. We further add to the corporate hypocrisy literature by reporting that the relationship between the temporal order of inconsistent CSR information and consumer cognitive dissonance is moderated by issue congruence between a CSR initiative and a social responsibility crisis, such that consumers perceive more cognitive dissonance when a company announces its CSR strategy first and then experiences a crisis than when the order is reversed, provided that the CSR announcement and the crisis are in the same domain. However, when a company experiences a crisis that is unrelated to its CSR initiative, consumers view the company as

moderately hypocritical regardless of whether the announcement or the crisis occurred first. These findings confirm an aggravating effect of congruence between CSR initiative and CSR crisis, as suggested by Kim and Choi [10], as well as by Effron and Monion [64]. Moreover, we identify perceived corporate hypocrisy as a meaningful mediator between cognitive dissonance and corporate reputation in addition to cognitive dissonance's direct detrimental effect on corporate reputation.

Finally, and contrary to our prediction, we found that consumers perceived a company with inconsistent words and practices as equally hypocritical whether the words were announced first or second. This could be because consumers view a crisis on one social issue and a CSR initiative on another as two separate issues and thus do not relate the consequences for one to the consequences for the other, which would limit cognitive dissonance and perceptions of corporate hypocrisy.

Our paper also provides important implications for marketing practitioners. As inconsistent CSR information triggers consumer cognitive dissonance and perceptions of corporate hypocrisy, which in turn tarnishes the corporate reputation. Therefore, marketing practitioners should be alert when implementing CSR strategies. We suggest that marketers should actively engage consumers as they initiate CSR campaigns to build corporate brand trust because brand trust not only reduces perceptions of corporate hypocrisy but also improves corporate reputation [29].

Another managerial implication of our study lies in the finding that when a company faced a crisis in the same domain as its CSR initiative, a proactive CSR strategy had a more detrimental effect on consumer perceptions than a reactive strategy did. However, the temporal order of the CSR initiative and CSR crisis had no significant impact on consumer perceptions when the initiative and the crisis were in unrelated domains. This finding suggests that companies that undertake CSR initiatives and communication should take precautions against potential crises, particularly in the same social domain as the initiative. Our findings support Kim and Choi's [10] suggestion that in the event of crises, decision-makers should initiate CSR campaigns in the same domain as the crisis because this reactive CSR strategy triggered less negative consumer perceptions.

### 5.2. Limitations and Future Research Directions

As with any study, we acknowledge several limitations that suggest directions for future research. First, we studied a fictitious company with a relatively small sample size, and thus, we question the external validity of our findings. Future researchers could employ a larger and more representative sample and study an existing company to address this issue.

Second, we did not consider the influence of the social issue relevance to consumers. We conducted our study because our experimental scenarios were designed with an environment issue and a fair wage issue, which are both relatively relevant to consumers. However, because individuals have strong feelings about such a wide range of social issues, researchers of future studies could design their experiment with social issues concerned with different levels of issue relevance and control it, and then examine the influence of inconsistent CSR information. Lastly, we showed a relationship direction from cognitive dissonance to corporate hypocrisy and then to corporate reputation, but reversed relationships among variables are also possible, especially when variables are related to psychological perceptions. Therefore, we suggest that more concrete and robust findings in the causal relationships among variables could be obtained in future research.

**Author Contributions:** J.X. and E.-K.H. conceived the initial research idea; J.X. performed the literature review, designed the experiments, and analyzed the data; J.X. both wrote and reviewed the manuscript. All authors have read and agreed to the published version of the manuscript.

**Funding:** This research received no external funding.

**Institutional Review Board Statement:** Not applicable.

**Informed Consent Statement:** Not applicable.

**Data Availability Statement:** The data presented in this study are available upon request from the corresponding author.

**Conflicts of Interest:** The authors declare no conflict of interest.

## Appendix A

Company H general information (a mock company, Company H):

Company H is a multinational clothing retail company well-known for fast-fashion products. It operates retail and manufacturing businesses around the world. The company also takes an active role in society.

Scenario 1: proactive (initiative first) × congruence:

Initiative:

Two weeks ago, Company H posted on its official website the fact that too many textiles in the fashion industry end up burned or in landfills, causing intense environmental pollution. As a member of the fashion industry and society as a whole, we are determined to use only recycled materials as raw materials by 2030, thereby reducing textile waste and carbon dioxide emissions to protect the environment.

Crisis issue:

Today, it was reported in a local newspaper that Company H burned tons of unsold clothes as well as recycled textiles in a local landfill. According to the newspaper, the burning of these textiles emitted a great amount of carbon dioxide into the air, greatly polluted the environment, and induced great outcry from local residents.

Scenario 2: proactive (initiative first) × incongruence

Initiative:

Two weeks ago, Company H posted on its official website the fact that too many textiles in the fashion industry end up burned or in landfills, causing intense environmental pollution. As a member of the fashion industry and society as a whole, we are determined to use only recycled materials as raw materials by 2030, thereby reducing textile waste and carbon dioxide emissions to protect the environment.

Crisis issue:

Today, it was reported in a local newspaper that one of Company H's factories in Southeast Asia has been paying workers lower than the minimum hourly wage set by the local government. According to the newspaper, workers in the factory have to work long hours each day and receive much lower hourly wages than the government mandates. Workers expressed great dissatisfaction with this state of affairs.

Scenario 3: reactive (crisis first) × congruence

Crisis issue:

Two weeks ago, it was reported in a local newspaper that Company H had burned tons of unsold clothes as well as recycled textiles in a local landfill. According to the newspaper, the burning of these textiles emitted a great amount of carbon dioxide into the air, greatly polluted the environment, and induced great outcry from local residents.

Initiative:

Today, Company H posted on its official website that too many textiles in the fashion industry end up burned or in landfills, causing heavy environmental pollution. As a member of the fashion industry and society as a whole, we are determined to use only recycled materials as raw materials by 2030, thereby reducing textile waste and carbon dioxide emissions to protect the environment.

Scenario 4: reactive (crisis first) × incongruence

Crisis issue:

Two weeks ago, it was reported in a local newspaper that one of Company H's factories in Southeast Asia has been paying workers lower than the minimum hourly wage set by the local government. According to the newspaper, workers in the factory have to work

long hours each day and receive much lower hourly wages than the government mandates. Workers expressed great dissatisfaction with this state of affairs.

Initiative:

Today, Company H posted on its official website that too many textiles in the fashion industry end up burned or in landfills, causing intense environmental pollution. As a member of the fashion industry and society as a whole, we are determined to use only recycled materials as raw materials by 2030, thereby reducing textile waste and carbon dioxide emissions to protect the environment.

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
