# Peer review of "How Temporal Order of Inconsistent CSR Information Affects Consumer Perceptions?"

_sustainability, doi:10.3390/su13084292_

Round 1

Reviewer 1 Report

The article is fairly straightforward and reasonably interesting.  I would like to see some discussion of effect heterogeneity.  Looking at your sample, you should be able to see if there are substantial male/female differences and education/income differences in your findings.

Reviewer 2 Report

This study investigates the moderating effects of issue congruence between a CSR initiative and a crisis in the link between the temporal order of a CSR initiative and a CSR crisis and perceived corporate hypocrisy.

This study finds:

  • Consumers experience greater cognitive dissonance and perceive more corporate hypocrisy when they are exposed first to a CRS initiative and then to a crisis;
  • No significant differences when the CSR initiative is incongruent with the crisis; and
  • Consumer cognitive dissonance influences directly and indirectly (through perceived corporate hypocrisy) perceived corporate reputation.

This study examines an interesting and relevant topic: the communication of CSR, perceived corporate hypocrisy and reputation. However, I see some points that can be improved in the paper, in terms of contribution and objectives/methodology.

My main comments and recommendations are:

  • Contribution

In the abstract, the authors write that the paper has two contributions: a theoretical one (better understanding of the consumer psychological process) and a managerial one (providing insight into corporate crisis). But in the introduction, the authors write that the contribution of this paper is to “better understanding of perceived corporate hypocrisy and offer practical implications for marketing practioners”. I did not understand this last contribution: are the authors suggesting that CSR communication should be used as a marketing tool? I think that the contribution of the paper should be clear and consistent between what is written in the abstract and in the introduction.

For me, it is also unclear how can we distinguish this paper (in terms of contribution) from Wagner et al. (2009)’ paper. The authors say that the difference is that Wagner et al. (2009) did not consider the congruence effect and in this paper the authors include that effect. However, the results show that there are no significant differences when the CSR initiative is incongruent with the crisis. So, why is it worthwhile studying the congruence versus incongruence between CSR initiatives and crisis?

  • Objectives/methodology

I think that the relevance of the CSR communication is important to analyse the CSR initiative, crisis, perceived corporate hypocrisy and reputation. In this study, the authors did not control for the relevance of the CSR communication. They present this as a limitation. But my question is: Is the relevance of CSR communication not an aspect that is too essential for it too have been excluded? If the answer is yes (it is relevant), why it was not considered (the authors should include an explanation). If the answer is no (it is not relevant), why is  it a limitation.

Other comments:

  • The title of the paper is very long.
